Characterization of DNA methylation in Malawian Mycobacterium tuberculosis clinical isolates

Ndhlovu Victor vndhlovu@medcol.mw 1 2 3
Kiran Anmol 4 5
Sloan Derek J. 6
Mandala Wilson 4 7
Nliwasa Marriott 1 3
Everett Dean B. 4 5
Kumwenda Benjamin 1
Mwapasa Mphatso 3
Kontogianni Konstantina 8
Kamdolozi Mercy 1
Corbett Elizabeth 3 4 9
Caws Maxine 8 10
Davies Gerry 2 4
1 University of Malawi, College of Medicine , Blantyre , Malawi
2 University of Liverpool , Liverpool , United Kingdom
3 Helse Nord Tuberculosis Initiative Project, University of Malawi, College of Medicine , Blantyre , Malawi
4 Malawi-Liverpool Welcome Trust , Blantyre , Malawi
5 University of Edinburgh , Edinburgh , United Kingdom
6 Department of Medicine, University of St. Andrews , St. Andrews , United Kingdom
7 Academy of Medical Sciences, Malawi University of Science and Technology , Thyolo , Malawi
8 Liverpool School of Tropical Medicine , Liverpool , United Kingdom
9 London School of Hygiene & Tropical Medicine, University of London , London , United Kingdom
10 Birat Nepal Medical Trust, Lazimpat, Kathmandu , Kathmandu , Nepal
Uversky Vladimir
Electronic publication date: 2020 Dec 16
Publication date: 2020
Volume: 8
Electronic Location ID: e10432
Received 2020 Aug 12; Accepted 2020 Nov 5
Copyright: ©2020 Ndhlovu et al.
Copyright year: 2020
Copyright holder: Ndhlovu et al.
License: This is an open access article distributed under the terms of the Creative Commons Attribution License, which permits unrestricted use, distribution, reproduction and adaptation in any medium and for any purpose provided that it is properly attributed. For attribution, the original author(s), title, publication source (PeerJ) and either DOI or URL of the article must be cited.
License URL: https://creativecommons.org/licenses/by/4.0/

Keywords: Mycobacterium tuberculosis, DNA methylation, Single Molecule Real Time Sequencing, Malawian Mtb clinical isolates, Motif

Funding: Helse Nord Tuberculosis Initiative (No.095) This work was supported by the Helse Nord Tuberculosis Initiative (No.095). The funders had no role in study design, data collection and analysis, decision to publish, or preparation of the manuscript.

==============================
Background

Although Mycobacterium tuberculosis (Mtb) strains exhibit genomic homology of >99%, there is considerable variation in the phenotype. The underlying mechanisms of phenotypic heterogeneity in Mtb are not well understood but epigenetic variation is thought to contribute. At present the methylome of Mtb has not been completely characterized.

Methods

We completed methylomes of 18 Mycobacterium tuberculosis (Mtb) clinical isolates from Malawi representing the largest number of Mtb genomes to be completed in a single study using Single Molecule Real Time (SMRT) sequencing to date.

Results

We replicate and confirm four methylation disrupting mutations in 4 lineages of Mtb. For the first time we report complete loss of methylation courtesy of C758T (S253L) mutation in the MamB gene of Indo-oceanic lineage of Mtb. Additionally, we report a novel missense mutation G454A (G152S) in the MamA gene of the Euro-American lineage which could potentially be attributed to total disruption of methylation in the CCCAG motif but partial loss in a partner motif. Through a genomic and methylome comparative analysis with a global sample of sixteen, we report previously unknown mutations affecting the pks15/1 locus in L6 isolates. We confirm that methylation in Mtb is lineage specific although some unresolved issues still remain.

Introduction

Mycobacterium tuberculosis (Mtb) is the causative agent of Tuberculosis (TB), infecting about 1.7 billion people with 5–10% disease development rate in their lifetime (WHO, 2020). Although, Mtb strains exhibit >99% genomic homology (Hershberg et al., 2008), virulence heterogeneity has been observed. The Beijing strain is associated with increasing multidrug resistant TB (MDR-TB) (Cowley et al., 2008; Van der Spuy et al., 2009) whereas the East African Indian (EAI) lineage has been associated with lower rates of transmission compared to other lineages (Albanna et al., 2011). Similarly, the Euro-American lineage is the most geographically successful strain (Gagneux & Small, 2007) but specific associated mechanisms supporting this success remain unknown. The Mtb genome consists of uniformly distributed high GC content (>60%) with minimal non-native DNA incarcerations. Approximately 10% of the total coding potential contains polymorphic guanine-cytosine repetitive sequences (PGRS) (Cole, 2002; Grover et al., 2018) which encode two unrelated families of acidic glycine-rich proteins- proline-glutamic acid (PE) and proline-proline glutamic acid (PPE). Specific functions of these genes and their proteins remain unclear. However, proteins have been reported in the cell wall and the cell membrane, playing roles in antigenic diversity, immune evasion and virulence (Fishbein et al., 2015; Phelan et al., 2016), leading to phenotypic differences. Phenotypic heterogeneity in Mtb has been associated with epigenetic inheritance (Balaban et al., 2004) and the most common epigenetic mechanism in Mtb is DNA methylation (Casadesus & Low, 2006; Shell et al., 2013). A few studies have characterized the Mtb methylome and revealed three 6-methyladenine (m6A) motifs and their cognate methyltransferases (Mtases), MamA, MamB and HsdM respectively (Shell et al., 2013; Zhu et al., 2015). Using Pacific Biosciences Single Molecule Real Time (SMRT) sequencing, two studies have recently shown that specific mutations in the Mtases lead to loss of Mtase activity and may play a role in evolution of Mtb (Zhu et al., 2015; Phelan et al., 2018). At present the methylome of Mtb has not been completely characterized, neither has any resulting information been correlated with phenotypic heterogeneity observed in TB patients. Understanding the complete biology of Mtb will aid in developing strategies for reducing the Mtb treatment duration from the standard 6 months.

We present characterization of methylomes of 18 Mtb isolates from patients in Blantyre, Malawi. This work presents the largest number of Mtb genomes from a single location to be completed in a single study using Single Molecule Real Time (SMRT) sequencing to date. We confirm three confident sequence motifs in Mtb including the strain specific mutations responsible for loss of Mtase activity in Mtb. Additionally, for the first time we report the complete loss of methylation courtesy of a novel mutation C758T (S253L) in Indo-oceanic lineage (L1). We further report of a novel missense mutation G454A (G152S) in the MamA gene of Euro-American lineage which could potentially be attributed to total disruption of methylation in the CTCCAG motif but partial loss in a partner motif. Through a genomic and methylome comparative analysis with a global sample of 16 samples we report previously unreported mutations affecting the pks15/1 locus in L6 isolates.

Materials and Methods

Sample collection

Frozen archived clinical isolates from a parent previous prospective cohort study, Studying Persistence and Understanding Tuberculosis in Malawi (S.P.U.T.U.M) (Sloan et al., 2015) for which ethical approval had been obtained from the College of Medicine Ethics Committee, University of Malawi (P.01/10/855) were characterized. These were from patients aged 16–65 years old presenting with bacteriologically culture confirmed pulmonary Mtb between June 2010 and December 2011 at Queen Elizabeth Central Hospital in Blantyre, Malawi from whom prior informed consent had been obtained under the SPUTUM study. Out of a total of 133 Mtb positive isolates, 18 were selected based on which isolates were the first to be successfully revived from frozen state and used in this study.

Bacterial growth conditions

All experiments involving Mtb were performed in a Biosafety Level (BSL) 3 Laboratory, University of Malawi-College of Medicine/ Malawi Liverpool Welcome Trust (CoM/MLW) TB laboratory and at Liverpool School of Tropical Medicine following Standard Operating Procedures (SOPs). All reagents used were from Sigma-Aldrich unless otherwise stated.

For liquid culture, strains were grown in Middlebrook 7H9 broth base supplemented with oleic acid, albumin, dextrose and catalase (OADC) and an antibiotic mixture of polymyxin B, amphotericin B, nalidixic acid, trimethoprim and azlocillin (PANTA). Tubes were incubated in a BACTEC MGIT 960 instrument at 37 °C and monitored once a week for possible growth for up to eight weeks. Isolates used in the study were from a previous study for which ethics approval had previously been granted by the College of Medicine Ethics Committee (COMREC), University of Malawi (Sloan et al., 2015). Solid culture inoculation was done on Lowenstein-Jensen (LJ) slopes following laboratory SOP. Cultures were grown to mid-log phase and harvested at ∼7th week and used for DNA isolation. Mtb was confirmed using both the BD MGIT TBC ID test device (Becton Dickinson, Maryland U.S.A) following manufacturer’s instructions and Ziehl Neelsen (ZN) staining for acid fast bacilli (AFB).

DNA extraction

Genomic DNA was isolated using the traditional Cetyltrimethylammonium bromide (CTAB) method as previously described (Somerville et al., 2005). Extracted DNA was quantified using Qubit 3.0 fluorometer (Life Technologies, USA) according to manufacturer’s instructions and DNA purity was determined on a NanoDrop ND-1000 Spectrophotometer V3.7 (Thermo Scientific, Wilmington U.S.A) following manufacturer’s instructions. DNA purity was checked at absorbance 260 nm and 280 nm by calculating a ratio of A260/A280. DNA quality was analyzed on 1.5% Agarose Gel electrophoresis and visualized under UV light following ethidium bromide staining.

Genotyping of Mtb Isolates

Genotyping of the isolates was done at the Liverpool School of Tropical Medicine, United Kingdom. Lineage specific deletions were detected using a singleplex PCR based method with specific oligonucleotide primers targeting the regions of difference RD239, RD105 and RD750. PCR reactions were performed as documented in our previous publication (Ndhlovu et al., 2019).

DNA sequencing

Purified genomic DNA libraries were sequenced at the Centre for Genomic Research (CGR), Institute of Integrative Biology, University of Liverpool, United Kingdom. DNA libraries were purified with 1x cleaned AMPure beads (Agencourt) and the quantity and quality was assessed using the Qubit and NanoDrop assays respectively. In addition, the Fragment Analyzer using a high sensitivity genomic kit (Advanced Analytical Technologies, Inc.) was used to determine the average size of the DNA and the extent of degradation. DNA was treated with Exonuclease V11 at 37 °C for 15 min. The ends of the DNA were repaired as described by the manufacturer (Pacific Biosciences, Menlo Park, CA, USA). The sample was incubated for 20 min at 37 °C with DNA damage repair mix supplied in the SMRTbell library kit (Pac Bio). This was followed by a 5-minute incubation at 25 °C with end repair mix. DNA was cleaned using 0.5×  AMPure and 70% ethanol washes. DNA was ligated to adapter overnight at 25 °C. Ligation was terminated by incubation at 65 °C for 10 min followed by exonuclease treatment for 1 h at 37 °C. The SMRTbell library was purified with 0.5x AMPure beads. The library was size selected with 0.75% blue pippin cassettes in the range 7,000–20,000 bp. The recovered fragments were damage repaired again. The quantity of library and therefore the recovery was determined by Qubit assay and the average fragment size determined by Fragment Analyzer. SMRTbell library was annealed to sequencing primer at values predetermined by the Binding Calculator (PacBio) and a complex made with the DNA polymerase (P6/C4 chemistry). The complex was bound to Magbeads and this was used to set up the required number of SMRT cells for the project (two for each sample). Sequencing was performed on Pacific Biosciences RSII sequencing system (Pacific Biosciences, Menlo Park, CA, USA) using 360-minute movie times per cell, yielding ∼300x average genome coverage.

Bioinformatics analysis

Generated long Pacbio reads were analyzed using the RS_Modification_and_Motif_Analysis.1 protocol as part of SMRT analysis in SMRT Portal (version 2.2.0). To increase the robustness of our analysis, we included previously published Mtb methylation study Pacbio data (Bioproject: PRJEB21888) (Phelan et al., 2018) and conducted both genomic and methylation comparisons of the two datasets. Although Bioproject PRJEB21888 had 18 genomes, we could only access 16 and these were used in our analysis. However, we evaluated PRJEB21888 sequences using SMRT Portal (version 5.1.0). Reads were mapped using the Basic Local Alignment with Successive Refinement (BLASR) (Chaisson & Tesler, 2012) algorithm within the SMRT portal. Strain specific genomes were generated by mapping the reads to the reference genome (H37Rv) using Quiver tool. Standard settings were used to detect base modifications and methylation motifs in the strain’s genome. Inter-pulse duration (IPD) ratio (observed vs expected) was measured for the modification detection (Zhu et al., 2015). Computational validation of our samples’ lineages and lineage identification of PRJEB21888 samples were done using TB-Profiler (Phelan et al., 0000). Comparative analysis of pks15 (Rv2947c) gene was used to report lineages of the samples specifically those from PRJEB21888. The MAFFT (version 7.310) (Katoh & Standley, 2013) was used to generate multiple sequence alignment of consensus sequences against H37Rv reference. Following removal of the reference from the alignment, maximum likelihood (ML) phylogeny was constructed for the remaining sequences using RaxML (v8.2.12) GTR+Γ model (Stamatakis, 2014) applying 1000 bootstrap iterations. Although Mtb has a highly rigid and non-recombinogenic genome (>99% nucleotide identity), to report diverse genomic regions among isolates, Gubbins (2.4.1) (Croucher et al., 2015) was applied with the default parameters over previously generated alignment of 34 genomes and earlier constructed ML phylogeny as an initial tree. Identified recombination hot spots were plotted with phylogeny generated without hot spots, affected genes details and the metadata using Phandango (Hadfield et al., 2018). Samples were clustered hierarchically based on m6A IPD ratio pattern.

Multiple sequence alignment of Mtase genes (mamA, mamB, hsdM and hsdS) sequences against the reference gene from H37Rv genome was used to identify potential mutations responsible for loss of methylation. Comparative analysis of well characterized methylation sites among samples were performed. Clustering of the samples based on their reported IPD ratios at methylated sites was performed and compared with clustering in ML phylogeny.

Results

Lineage analysis of Mycobacterium tuberculosis

Experimental (RD-PCR) and computational (TB-Profiler) outcomes on Malawian strains, lineage identification was consistent revealing 3/18 (17%) L1 (Indo-Oceanic), 3/18 (17%) L2 (East-Asian) and 12/18 (66%) L4 (Euro-American). De novo reporting of global sample lineages (Phelan et al., 2018) (16 samples) using TB-Profiler revealed 3/16 L1(Indo-oceanic), 2/16 L2 (East-Asian), 3/16 L4 (Euro-American), 2/16 L5 (West African 1 and 6/16 L6 (West African 2) (Table 1). Using intact pks15 (Rv2947c) gene, it was possible to identify the 15/34 strains belonging to L4 in the combined dataset with a 7 bp deletion (GGGCCGC) in the pks15/1 gene as previously documented (Constant et al., 2002; Gagneux & Small, 2007). Additionally, pks15 (Rv2947c) could be used to assign lineages to the rest of the samples. All L1 (6/34 strains) had a G1318C substitution and GGGCCGC insertion while L2 (5/34) strains had a GGGCCGC insertion only. All L5 samples had a 9 bp deletion (CGGTGCTGG,1097-1105), a distinct substitution A50G and an insertion GGGCCGC. A L1, L5, L6 (G1318C substitution) and a L6 (1658 1 bp insertion of G), L1, L2, L5 (1658 7 bp insertion) (Fig. 1).

Table 1 Lineages and sub-lineages of the samples reported by TB-Profiler using assembled genomic sequences (ERS-Malawian and SAMEA-global samples).

Sample_ID	Lineage	Sub-lineage	Sub-sub-lineage	
ERS2711939	Lineage4	Lineage4.3	Lineage4.3.4	
ERS2711940	Lineage4	Lineage4.3	Lineage4.3.4	
ERS2711941	Lineage4	Lineage4.3		
ERS2711942	Lineage4	Lineage4.3	Lineage4.3.4	
ERS2711943	Lineage1	Lineage1.1	Lineage1.1.3	
ERS2711944	Lineage4	Lineage4.3	Lineage4.3.4	
ERS2711945	Lineage4	Lineage4.3	Lineage4.3.4	
ERS2711946	Lineage4	Lineage4.3	Lineage4.3.4	
ERS2711947	Lineage4	Lineage4.3	Lineage4.3.4	
ERS2711948	Lineage1	Lineage1.1	Lineage1.1.3	
ERS2711949	Lineage4	Lineage4.3	Lineage4.3.4	
ERS2711950	Lineage4	Lineage4.5		
ERS2711951	Lineage4		Lineage4.1.2	
ERS2711952	Lineage4	Lineage4.3	Lineage4.3.4	
ERS2711953	Lineage2	Lineage2.2		
ERS2711954	Lineage2	Lineage2.2		
ERS2711955	Lineage2	Lineage2.2		
ERS2711956	Lineage1	Lineage1.1	Lineage1.1.3	
SAMEA104606019	Lineage1	Lineage1.1	Lineage1.1.3	
SAMEA104606020	Lineage1	Lineage1.1	Lineage1.1.3	
SAMEA104606021	Lineage1	Lineage1.1	Lineage1.1.3	
SAMEA104606022	Lineage5			
SAMEA104606023	Lineage2	Lineage2.2	Lineage2.2.1	
SAMEA104606024	Lineage4	Lineage4.3	Lineage4.3.4	
SAMEA104606025	Lineage4	Lineage4.1	Lineage4.1.2	
SAMEA104606026	Lineage6			
SAMEA104606027	Lineage6			
SAMEA104606028	Lineage2	Lineage2.2	Lineage2.2.1	
SAMEA104606029	Lineage4	Lineage4.3	Lineage4.3.4	
SAMEA104606030	Lineage5			
SAMEA104606031	Lineage6			
SAMEA104606032	Lineage6			
SAMEA104606033	Lineage6			
SAMEA104606034	Lineage6			

Figure 1 Lineage specifi.c sequence variants relative to the reference (H37Rv) gene pks15 (Rv2947c).

The pks15 gene from 34 samples was aligned against the reference (H37Rv) to display the differences at four locations/ranges within the gene, discriminating four lineages L5 (A50G substitution), L5 (1097-1105 CGGTGCTGG deletion), L1, L5, L6 (G1318C substitution) and L6 (1658 1 bp insertion of G), L1, L2, L5 (1658 7bp insertion).

DNA methylation patterns

The m6A motifs present in more than 10 isolates were CACGCAG (820 sites), CTCCAG (1947 sites), CTGGAG (1947 sites), GATN4RTAC (363 sites) and GTAYN4ATC (363 sites), Motifs CTCCAG and GATN4RTAC are paired with CTGGAG and GTAYN4ATC respectively (Table S1). All L2 samples and one L6 sample (SAMEA104606027) lacked methylation in CTGGAG/CTCCAG motif. One sample from L4 (ERS2711941) lacked methylation in the CTCCAG motif (Fig. 2A). The CTCCAG motif is methylated by mamA Mtase (Shell et al., 2013). This L4 sample had a synonymous substitution at C216T and a non-synonymous substitution at G454A resulting in G152S amino acid substitution in the MamA gene. To our knowledge, this potentially methylation disrupting mutation has not been previously reported. Multiple sequence comparison with the reference gene (Rv3263) revealed all L2 samples as possessing a A809C (E270A) change as previously reported (Shell et al., 2013; Zhu et al., 2015; Phelan et al., 2018) (including G1199C, W400S in only two samples). Consequently, all non-methylated L6 samples possessed a A1378G (A460T) mutation in Rv3263 gene. The CACGCAG motif is methylated by the mamB Mtase (Zhu et al., 2015; Phelan et al., 2018) and two of the six L1 samples (ERS2711948, ERS2711956) lacked the mamB Mtase (Fig. 2A). Methylation in the rest of the samples was however below 80% (range 56% - 79.6%) (Fig. 2C). Interestingly, all L1 samples (6) possessed a C758T (S253L) mutation in the mamB gene (Rv2024c). This mutation has previously been reported to be associated with partial loss of Mtase activity in L1 samples (Phelan et al., 2018). This is the first time that mutation S253L is being associated with complete loss of MamB Mtase function. Lineage 2 (ERS2711953) and L4 (ERS2711945) samples possessed low methylation in CACGCAG motif (65% and 73% respectively) compared to other samples from the same lineage (100%) despite absence of any specific mutations in the mamB gene. Lineage specific mutation R289C (L6) and L452V (L5) did not affect the mamB Mtase activity (Fig. 2C). However, non-lineage specific multiple variation was reported at 3′  end. Motifs GATN4RTAC (363 sites) and GTAYN4ATC (363 sites) are methylated by hsdM (Rv2756c) and hsdS (Rv2761c) genes (Zhu et al., 2015; Phelan et al., 2018). Although one L1 sample (ERS2711956) lacked methylation in either motif the other, (ERS2711948) was methylated at GTAY N4ATC only (Fig. 2A). The hsdM gene sequences were identical for all L1 samples and no 5′-upstream alterations (300 bp) were reported. All the L4 samples lacking GATN4RTAC/GTAYN4ATC methylation had mutations at T917C resulting in L306P amino acid change in hsdM gene and G74T resulting in G25V amino acid change in hsdS gene. While the T917C (L306P) mutation was previously associated with loss of methylation (Shell et al., 2013; Zhu et al., 2015; Phelan et al., 2018), the G74T(G25V) mutation in hsdS gene has not been previously reported. The distribution of lineage specific motif methylation is shown in Fig. 2B and Fig. 2D. High variability in methylated motifs was observed in “modern” strains (L2/L4) defined by a conserved TbD1 genomic region while methylation was largely uniform in “ancient” strains (L5/L6) with a deleted TbD1 genomic region, with all the detected motifs being methylated (Figs. 2A and 2B). However, L1 (an ancient strain) was found to have the greatest variability of all (standard deviation >0.2 with the rest of the samples being <0.2). The high variability in L1 was a result of a few global samples clustering separately as opposed to the Malawian samples. In summary L4 isolates were methylated at a higher frequency for all three motifs than the rest of the isolates (Figs. 2A and 2B) and a higher fraction of modified motifs (Figs. 2C and 2D).

Figure 2 Methylation summary.

(A) Distribution of methylated samples in each lineage for the motifs. (B) Distribution of samples with methylated motifs in each lineage. (C) Methylation efficiency in samples for each motif. (D) Methylation efficiency by motif in each lineage (numbers in parenthesis after the motif represent total number of sites with that motif).

Figure 3 Tanglegram of hierarchically clustered samples.

Clustering was based on IPD and ML phylogeny. Samples are coloured based on lineages. Three samples clustered separately from their lineage. Two L4 samples (ERS2711951 and SAMEA104606025) with L6 and L5 samples, respectively.

Methylation efficiency among lineages

Among the samples having methylation in major motifs, most reported methylation efficiency of higher than 82% (Fig. 2C, Table S1). Lineage 4 sample (ERS2711941) reported 58% methylation for CTGGAG motif but lacked methylation on the partner motif CTCCAG while another L4 sample (ERS2711945) reported 69% and 79% methylation on CTGGAG and CTCCAG respectively. Two L1 samples (ERS2711948, ERS2711956) reported methylation of 27% and 36%, 43% and 51% for CTGGAG and CTCCAG respectively but having no specific mutation in methylation conferring genes. Methylation distribution within motifs for each sample is displayed in Fig. 2D. Lineage 1 sample (ERS2711948) was methylated at 32% on motif G TAYN4ATC, while L2 sample (ERS2711953) was methylated at 48% and 49% on GATN4RTAC motif. Other samples with low efficiency were as follows: ERS2711953 from L2 with 65% methylation efficiency and L1 samples SAMEA104606020, SAMEA104606019, SAMEA104606021 and ERS2711943 with 71%, 80%, 75% and 56% respectively on CACGCAG motif (Fig. 2C and Fig. 2D). Interestingly, we found association between low methylation efficiency for all motifs (in 4/18 samples) and low sequencing coverage <50x compared to none in the rest >100x probably resulting from poor sample quality.

Comparison of methylation within Mtb strains

The strain arrangements in the m6A IPD ratio based cladogram clusters and genome based maximum likelihood (ML) phylogeny were compared (Fig. 3). The samples clustered into four IPD based groups. However, in the ML phylogeny lineages formed distinct clusters. One L6 sample (SAMEA104606027) having no CTGGAG/CTCCAG Mtase activity clustered as a L2 sample (Fig. 3). Two L4 samples (ERS2711951 and SAMEA104606025) lacking methylation in GATN4RTAC and GTAYN4ATC motifs clustered as L6 samples respectively (Fig. 3). Interestingly, the two L4 samples formed their own distinct cluster on ML phylogenetic tree.

In the recombination hotspot analysis, 256 genes were reported to have been affected (Fig. 4). Ninety-one well annotated genes were affected due to insertions/deletions (Indels) varying in size from 1 to 36016pb affecting a large number of PPE family (21), PE-PGRS family (25) and ESAT 6 (6) genes. Lineage specific recombination relative rate to mutation ratio (r/m) reported as L1: 0.968310, L2:1.865780, L4:4.915385, L5:1.001656, L6:1.066062.

Figure 4 Diversity of regions in different samples and lineages.

displayed in alignment frame of the dierent samples and lineages calculated with default Gubbins parameters. Regions of aected gene locations in the alignment, the phylogeny of the 34 samples and the recombination events.

Stability of methylation within mycobacterium tuberculosis strains

It was important to investigate the effect of culture media on methylation patterns. Two isolates (ERS2711943 and ERS2711952) were MGIT grown but their methylation patterns did not appear distinct from samples of the same lineage on solid culture except that “CACGCAG” motif for ERS2711943 reported the lowest methylation in the L1 samples and GATN4RTAC motif was detected as methylated in ERS2711943 only. No significant difference could be established between liquid and solid culture isolates for methylated motif CTGGAG (Fishers exact test p = 0.76). As for motif CACGCAG solid cultured isolates were methylated at an average 76% while liquid cultures were methylated an average 97%. It was found that liquid cultured isolates were significantly more methylated than solid cultures (Fisher’s exact p = 0.02) for motif CACGCAG. It was observed that this difference was as a result of sample ERS2711943 having low levels of methylation (56%) compared to the other samples at >95%.

We next investigated methylation within the gene regions and promoter regions of genes.

Methylation within gene regions ranged between 49% to 51% in each strand and there was no over representation of methylation by strand (Chi squared test with Yates correction P = 0.44). On the other hand, methylation within promoter regions of genes ranged from 37% to 62% by strand of the promoter methylation. Again there was no significant statistical differences observed by strand (Fishers exact test P = 0.19) suggesting growth media did not affect Mtase activity.

Discussion

Studies of Mtb DNA methylation using SMRT sequencing have previously focused on strains originating from Hongkong (2) (Leung et al., 2017) L2, China (12) (Zhu et al., 2015),L2–L4, L6, L8 and more recently a global sample that included Europe, Asia, West Africa and South Africa (16) (Phelan et al., 2018), L1-L2, and L4-L6. In our study the activity of Mtase could be inactivated by four different mutations somewhat in a lineage specific manner. Loss of MamA Mtase in L2 (Beijing) isolates was associated with a point mutation A809C (E270A) reported previously (Shell et al., 2013). Interestingly, L2 (Beijing) strains have a higher propensity to cause active disease and have been associated with increasing drug resistance in some geographical areas (Cowley et al., 2008); (Van der Spuy et al., 2009). A recent study however failed to establish a possible role of methylation in virulence of Beijing strains (“Computational characterisation of DNA methylomes in mycobacterium tuberculosis Beijing hyper- and hypo-virulent strains”). Similarly, absence of MamB Mtase in two (L1) Indo-oceanic isolates could be attributed to a C758T (S253L) novel missense mutation recently characterized elsewhere (Phelan et al., 2018) and confirmed in this study. While this mutation was putatively found to lead to partial methylation (50–60%) in a previous study, for the first time, we report that it could also lead to complete loss of Mtase activity as two of our L1 isolates lacked methylation. It is still unknown whether the C758T (S253L) mutation contributes to recent transmission of EAI6 family in L1(Duarte et al., 2017). Lack of HsdM methylation in L4 could be attributed to the C917T (P306L) mutation which was present in 11/12 Malawian isolates. These results are consistent with previous studies which seem to suggest that the P306L mutation is very common in L4 strains (Shell et al., 2013; Zhu et al., 2015). Lineage 4 isolates have the highest global prevalence than any other lineage and more studies will be required to establish whether loss of HsdM methylation could be associated with this global success. If indeed HsdM Mtase is disrupted by this mutation in L4, it remains intriguing how some L1 isolates could lose HsdM Mtase in absence of any mutation in the hsdM gene. The novel missense mutation G454A (G152S) could be attributed to total disruption of methylation in the CTCCAG motif and partial loss in the partner motif CTGGAG. Overall the higher occurrence of methylation disrupting mutations in L2 and L4 (modern strains) compared to L1, L5 and L6 (ancient strains) is suggestive of an evolutionary adaptation of these strains. The pks15/1 locus appears to distinguishes L5 isolates using 9 bp (CGGTGCTGG) deletion, a distinct substitution A50G and an insertion GGGCCGC while L6 isolates could also be classified using a 6bp (GGGCCGC). The pks15/1 locus therefore a potential valuable marker for identifying L5 and L6 isolates. These two lineages are localized to west Africa and these distinct genetic markers could help explain this regional restriction. The fact that our sequenced isolates were selected based on the first to be revived from frozen state may have resulted in a sampling bias of selecting a subpopulation with a particular methylation pattern. It has previously been demonstrated that methylation plays a key role in the growth of Mtb under discrete microenvironments (Shell et al., 2013). Based on this, our results on methylation patterns should be taken with caution as we have not included those samples that revive slowly. The large number of genomic re-arrangements observed in cell wall component genes PPE, PE-PGRS and ESAT-6 is evidence of the large variations that could potentially explain strain/lineage specific propensity to cause and transmit disease. Additionally, the high lineage specific recombination relative rate to mutation ratio observed in “ancient” strains (L1, L5 and L6) of ∼1 compared to “modern” strains (L2 and 4) of >2 may explain adaptation of strains to specific host populations and propensity to cause active disease.

Complete characterization of DNA methylation in Mtb could provide clues to some of the clinical phenotypes which have been associated with strain and lineage variation. Overall this study shows the potential of SMRT sequencing long reads to aid in better understanding the complete biology of Mycobacterium tuberculosis by resolving repetitive regions of the genome and elucidating the complete methylome of the pathogen. The high frequency of Mtase disrupting mutations in Mtb could point to a competitive fitness advantage such as immune evasion or even persistence. These sequences from Malawi could serve as references in future Mtb methylome studies. This study is limited by the fact that we could not establish via experimentation the direct association between specific mutations and loss of Mtase activity. To better understand the complete impact of DNA methylation within specific strains and lineages, subsequent studies will need to integrate transcriptomic and proteomic data to methylomes.

Conclusions

We conclude that DNA methylation is lineage specific although some unresolved issues still remain. Loss or absence of methylation may be used as a form of competitive fitness advantage by clinical Mycobacterium tuberculosis isolates.

Supplemental Information

Supplemental Information 1 Methylation efficiency for 34 Mycobacterium tuberculosis samples

Click here for additional data file.

We thank the guardians and patients who participated in this study, and the staff at Queen Elizabeth Central Hospital for their assistance.

Additional Information and Declarations

Competing Interests

Author Contributions

Data Availability

The authors declare there are no competing interests.

Victor Ndhlovu conceived and designed the experiments, performed the experiments, prepared figures and/or tables, authored or reviewed drafts of the paper, and approved the final draft.

Anmol Kiran performed the experiments, analyzed the data, prepared figures and/or tables, authored or reviewed drafts of the paper, and approved the final draft.

Derek J. Sloan, Wilson Mandala, Marriott Nliwasa, Dean B. Everett, Benjamin Kumwenda and Mphatso Mwapasa analyzed the data, authored or reviewed drafts of the paper, and approved the final draft.

Konstantina Kontogianni and Mercy Kamdolozi performed the experiments, authored or reviewed drafts of the paper, and approved the final draft.

Elizabeth Corbett analyzed the data, prepared figures and/or tables, and approved the final draft.

Maxine Caws conceived and designed the experiments, performed the experiments, prepared figures and/or tables, and approved the final draft.

Gerry Davies conceived and designed the experiments, performed the experiments, analyzed the data, prepared figures and/or tables, authored or reviewed drafts of the paper, and approved the final draft.

The following information was supplied regarding data availability:

Data is available at ENA: PRJEB28592.

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
