# Peer review of "Characterization of DNA methylation in Malawian Mycobacterium tuberculosis clinical isolates"

_PeerJ, doi:10.7717/peerj.10432_

## Round 0.1 · original submission · Minor Revisions

Please address all the issues pointed out by the reviewers and revise the manuscript accordingly.

Reviewer 1 ·

Basic reporting

No comment

Experimental design

No comment

Validity of the findings

No comment

Additional comments

Characterization of DNA methylation in Malawian Mycobacterium tuberculosis clinical isolates

Ndhlovu et al. report data from an interesting study examining potential mechanisms for MTB phenotypic heterogeneity, despite >99% genetic homology. Epigenetic factors play an important role in regulating an extensive number of processes and influencing phenotype. Since there is sparse data on the effects of epigenetic modifications on the phenotype of MTB this study is important to field.

Methods

1) Choice of isolates to study – lines 100-101 – “Out of a total of 133 MTB positive isolates, 18 were selected based on which isolates were the first to be successfully revived from frozen state”

Since this is an observational study the data is valid but please can the authors comment in the discussion on this strategy for selecting isolates – might speed of growth be related to epigenetic modifications. A comment should be made in the discussion as to how selection of isolates may have influenced the methylation data.

Results

2) Figure 1: Is the reference here from H37Rv? Please mention in the legend.

3) Figure 2: Please can you provide X and Y axis legends. Fig 2A and B particularly their Y axis is given as ‘0’. Please can you increase the font size of all text on Figure 2 – it is very hard to read the legends and labels. I don’t find this figure very easy to interpret.

You are reporting ‘Distribution of methylated samples in each lineage for the motifs. It might help if there were numbers stated in the legend or on the bars. How many samples of each lineage do you have here? Maybe it is because the Y axis doesn’t have a legend that I cannot quite follow this.

Fig 2 C and D – You state that there is high variability in methylated motifs in modern strains (L2/L4) while largely uniform in ancient strains (L1/L5/L6).

Please can you state how ‘modern’ and ‘ancient’ are defined.

In Fig2D – L5 and L6 indeed appear to have little variation – the points are all around 1.0 in the violin plot – however L1 looks to have the largest variation of all. Please can you explain why you consider this variation less than L2/L4.

4) You state in line 266 that ‘in some cases, low methylation could possibly be attributed to sequencing errors”. This is important and you should give a numerical range or expected inaccuracy of the results based on system used.

5) Line 294 “ as a result of sample ERS2711943 being lowly methylated at 56%”

Please change to “ as a result of sample ERS2711943 having low levels of methylation (56%) as compared to the other samples at >95%.

6) Line 46-47 “We report previously unreported mutations…”
Please change to “We report previously unknown mutations”

Discussion

7) ‘Studies of Mtb DNA methylation using SMRT sequencing have previously focused on strains originating from the USA, Asia and…’ please can you give approx. numbers studied and lineage to put your study better into context.

Reviewer 2 ·

Basic reporting

The manuscript by Ndhlovu et al provides insights into the methylation profile of clinical strains of Mycobacterium tuberculosis from Malawi. This study provides evidence of methylation events and lineage specific comparisons of methylation activities of a total of 18 strains. As methylation profiling of bacteria with clinical importance such as M. tb is a new field made possible with the current advances in whole genome sequencing this study paves the way for future more advanced studies to follow where specific methylation signatures could be mapped to specific phenotypic traits.

Experimental design

This manuscript is clearly written, provide enough background and referencing relevant papers in the field. Hypothesis and experimental approaches are well documented with all ethical standards.

Validity of the findings

The data analysis is robust, and the authors highlight the importance of novel mutations found in methyltransferases in M. tuberculosis. The authors present their data clearly in figures 1 to 3. However, figure 4 would have been greatly improved by presenting the different strains in a circular representation as it is common when you compared variability of DNA.

External reviews were received for this submission. These reviews were used by the Editor when they made their decision, and can be downloaded below.

---

## Round 0.2 · accepted · Accept

Thank you for addressing all the critiques of the reviewers. I am pleased to accept your revised manuscript.

External reviews were received for this submission. These reviews were used by the Editor when they made their decision, and can be downloaded below.